

# Experimental input from $e^+e^-$ for $a_\mu$ light-by-light

### Yuping Guo[1][⋆] on behalf of BESIII Collaboration

**1** Institut für Kernphysik, Johannes Gutenberg-Universität Mainz, Mainz, Germany

⋆ guo@uni-mainz.de

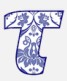

*Proceedings for the 15th International Workshop on Tau Lepton Physics,
Amsterdam, The Netherlands, 24-28 September 2018*

## Abstract

The anomalous magnetic momentum of the muon, $a_\mu$, has been measured and calculated with a precision up to 0.5 ppm, but there is a 3 to 4 standard deviations between these two values. The uncertainty in the calculation is dominated by the hadronic part, including the hadronic vacuum polarization and the hadronic light-by-light. The meson transition form factors and the helicity amplitudes can be used as input or constraint to the calculation of the hadronic light-by-light contribution. Latest experimental studies of the transition form factors of $\pi^0$, $\eta$, and $\eta'$ and the cross-section of $\gamma\gamma^* \to \pi^+\pi^-$ from $e^+e^-$ collider are presented.

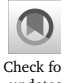
# 1 Introduction

The anomalous magnetic momentum of the muon, $a_\mu \equiv (g-2)$, has been considered as one of the observables with which the completeness of the Standard Model (SM) can be tested. The direct measurement from the BNL experiment yields $(11659208.9 \pm 6.3) \times 10^{-10}$, with a precision of 0.54 ppm [1]. The theoretical calculation in the SM has a similar precision [2–4]. The difference between the measurement and the calculation is 3 to 4 standard deviations. A new experiment, started in 2017 at Fermilab [5], as well as the planned experiment at J-PARC [6], aims to reduce the uncertainty of the direct measurement by a factor of four; an improvement of the SM prediction is urgently needed. The SM prediction contains the QED contribution, the weak contribution and the hadronic contribution. The QED contribution is the largest one, and has been calculated up to 5-loop in perturbation theory with a precision of 0.0007 ppm [7]. The weak contribution is small, has been calculated to 2-loop, with the measured Higgs mass taken into account [8], and its uncertainty is well under control.

The hadronic contribution is the second largest one, but the largest to the uncertainty of the SM calculation. It contains two components, the hadronic vacuum polarization (HVP) contribution and the hadronic light-by-light (HLbL) contribution. Although the absolute value of the HLbL is only 1.5% of the HVP, their uncertainties are at the same level. Improvements from both are needed. The calculation of the HVP contribution can be related to hadronic cross-sections via a dispersion relation, thus improving the accuracy of the cross-section measurement can directly improve the precision of the HVP calculation. Meanwhile the situation for the HLbL part is different. So far, there are only calculations from hadronic models. The validation of these models usually is done with the meson transition form factor (TFF). Although different models use the same data as constraint, the central values are different. Moreover, there is no reliable method to estimate the uncertainty of these models. Recently, data-driven dispersive approaches have been developed by two independent groups [9–16]. By using the meson TFF and the helicity amplitudes of the two-photon cross-section as input, the dispersive approaches build a direct relation between the HLbL contribution and experimentally measurable variables. It allows a more precise prediction of both the central value and the uncertainty. The dominant contribution from the HLbL comes from the pseudoscalar meson exchange, followed by the meson loop contribution. These input variables can be measured in the time-like regime through the meson Dalitz decay process or radiative process from $e^+e^-$ annihilation, or in the space-like regime through two-photon fusion process at $e^+e^-$ machine.

# 2 The BESIII experiment

The BESIII detector is a magnetic spectrometer [17] located at the Beijing Electron Positron Collider (BEPCII). The cylindrical core of the BESIII detector consists of a helium-based multilayer drift chamber (MDC), a plastic scintillator time-of-flight system (TOF), and a CsI(Tl) electromagnetic calorimeter (EMC), all enclosed in a superconducting solenoidal magnet providing a 1.0 T magnetic field. The solenoid is supported by an octagonal flux-return yoke with resistive plate counter muon identifier modules (MUC) interleaved with steel. The acceptance of charged particles and photons is 93% over $4\pi$ solid angle. The charged-particle momentum resolution at 1 GeV/$c$ is 0.5%, and the $dE/dx$ resolution is 6% for electrons from Bhabha scattering. The EMC measures photon energies with a resolution of 2.5% (5%) at 1 GeV in the barrel (end cap) region. The time resolution of the TOF barrel part is 68 ps, while that of the end cap is 110 ps. The position resolution in MUC is about 2 cm.

The BEPCII is a $\tau$-charm factory that works with center-of-mass (CM) energy from 2.0 to 4.6 GeV. The designed luminosity is $1 \times 10^{33}$ cm$^{-2}$s$^{-1}$. From 2009, the BESIII experiment

has collected large data samples at the full CM energies coverage region, including $5.9 \times 10^9$ events at the $J/\psi$ peak, $448.1 \times 10^6$ events at the $\psi(2S)$ peak, 2.9 fb$^{-1}$ at the $\psi(3770)$ peak, more than 15 fb$^{-1}$ at CM energies above 4.0 GeV, and a set of data samples at 151 CM energies covers the whole energy region used for measurements of $R$, $\tau$ physics, and baryon form factor measurement.

## 3   Measurement at $e^+e^-$ machine

The meson TFFs and helicity amplitude can be measured in space-like regime by using the two-photon fusion process at $e^+e^-$ machine or in time-like regime by using the Dalitz decay process. Figure 1 shows the tree-level Feynman diagram for the two-photon process, where $q_1$ and $q_2$ represents the momentum of the two photons emitted from the lepton lines. Three

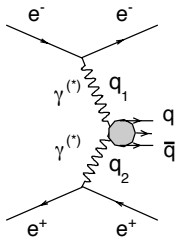

Figure 1: The Feynman diagram for the two-photon fusion process.

techniques are used to study the two-photon process depending on the number of leptons detected in the detector, namely, the untag, the single-tag, and the double-tag method. In the untag case, only the hadronic productions is detected, the directions of the leptons in the final state is required to parallel the beam direction. In this way, the virtuality of both photons is very small ($q_{1,\,2}^2 \simeq 0$), and can be considered as quasi-real. In the single-tag case, one of the leptons is detected in the detector, while the other is required to be scattered along the beam direction. In this case, the photon emitted from the tagged lepton is far off-shell, while the untagged one is quasi-real. The TFF as a function of $Q^2$, $F_{M\gamma^*\gamma^*}(q_1^2, q_2^2) \equiv F_{M\gamma^*\gamma}(Q^2)$ can be measured. In the double-tag case, all the particles in the final state are detected, the TFF $F_{M\gamma^*\gamma^*}(q_1^2, q_2^2)$ is accessible. This is the input variable which can be used directly in the dispersive approaches. The double-tag method is limited by statistics as the cross-section of the two-photon process strongly peaks at small angle, so most of the current measurements are done with untag or single-tag method. The studies presented here in space-like region are all performed in single-tag method.

## 4   Transition form factor measurement of pseudoscalar meson

The dominate contribution from the HLbL to $a_\mu$ comes from the neutral pseudoscalar exchange contribution, $\pi^0$, $\eta$, and $\eta'$ (see references from Ref. [2, 4]). Using a dispersive approach, the pseudoscalar contribution to $a_\mu^{\text{HLbL}}$ has been evaluated [23]. It can be factorized as a two-dimensional integral of the universal weight functions times the form factor dependent functions. The weight functions are model-independent. The study shows that the region of photon momenta below 1.0 GeV (1.5 GeV) for $\pi^0$ ($\eta$ and $\eta'$) gives the main contribution. The TFFs of these mesons in the space-like region have been measured by the BaBar [18,

19] and Belle [20] experiments recently, and in 1990s from the CELLO [21] and CLEO [22] experiments. The results from these experiments are shown in Fig. 2. The measurements from B-factories have high precision for $Q^2 \geq 4$ GeV$^2$. The CLEO measurement measures from $Q^2 \geq 1.5$ GeV$^2$. In the region with $Q^2 \leq 1.5$GeV$^2$, which is the most important region for $a_\mu^{\mathrm{HLbL}}$, the only measurement comes from the CELLO experiment with poor accuracy.

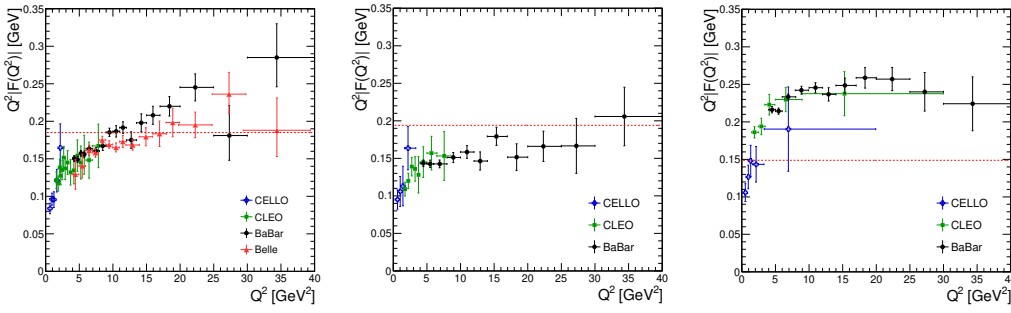

Figure 2: The TFF of $\pi^0$ (left), $\eta$ (middle), and $\eta'$ (right) measured from the CELLO [21], CLEO [22], BaBar [18, 19], and Belle [20] experiments.

### 4.1 Space-like transition form factor measurements

Comparing to the B-factories, the BESIII experiment runs at much lower CM energies, thus can measure the TFF in the lower $Q^2$ region. The data sample collected at the $\psi(3770)$ peak has been used to measure the TFFs of the $\pi^0$, $\eta$, and $\eta'$.

In the measurement of the TFF of $\pi^0$, the $\pi^0$ is reconstructed using its $\gamma\gamma$ final state. Events with only one lepton, two to four photons reconstructed in the detector are considered as the signal candidates. Using momentum conservation, the untagged lepton is required to fly along the beam direction, $|\cos\theta_{\mathrm{miss}}| \leq 0.99$. Background events mainly come from the radiative Bhabha scattering process, where the hard radiative photon combined with soft photons forms a fake $\pi^0$. These events has been suppressed with conditions put on the helicity angle of the $\pi^0$ candidates ($|\cos\theta_{\mathrm{H}}| \leq 0.8$). A further requirement of $\frac{\sqrt{s}-E^*_{l\pi^0}-p^*_{l\pi^0}}{\sqrt{s}} < 0.05$ is applied, where $E^*_{l\pi^0}$ and $p^*_{l\pi^0}$ are the sum of the energy and three-momentum of the tagged lepton and $\pi^0$ in the CM frame. This requirement suppresses events with large initial state radiation, leading to incorrect reconstruction of $Q^2$. The background events from charmonium decays with various hadrons in the final states can also be removed with this requirement. Events after these selections show a clear $\pi^0$ peak in the $\gamma\gamma$ invariant mass spectrum, as shown in Fig. 3. In the plots, the red histogram is from a signal Monte Carlo (MC) simulation by using EKHARA event generator [24, 25], other colored histograms are from background MC simulations. The discrepancy between data and MC simulations comes from the missing components in the MC simulations, which are the small angle Bhabha scattering events and the $f_2(1270)$ resonant from $\gamma\gamma \to \pi^0\pi^0$ process. The $Q^2$ from data and MC simulations are also shown in Fig. 3, the accessible $Q^2$ region is 0.3 GeV$^2$ to 3.1 GeV$^2$.

As the background events distributed smoothly along the $\gamma\gamma$ invariant mass distribution, the number of $\pi^0$ events is extracted by performing fits to the $\gamma\gamma$ invariant mass distributions in bins of $Q^2$. The fit is performed with a polynomial function in the $\pi^0$ sideband regions. The fitted curve is extrapolated to the $\pi^0$ signal region, the events above the extrapolated curve are considered as signal events. The sideband regions are defined as [0.070, 0.115] GeV/$c^2$ and [0.151, 0.200] GeV/$c^2$. With the reconstruction efficiency obtained from the signal MC simulation and the luminosity of the data sample, the differential cross section $d\sigma/dQ^2$ is

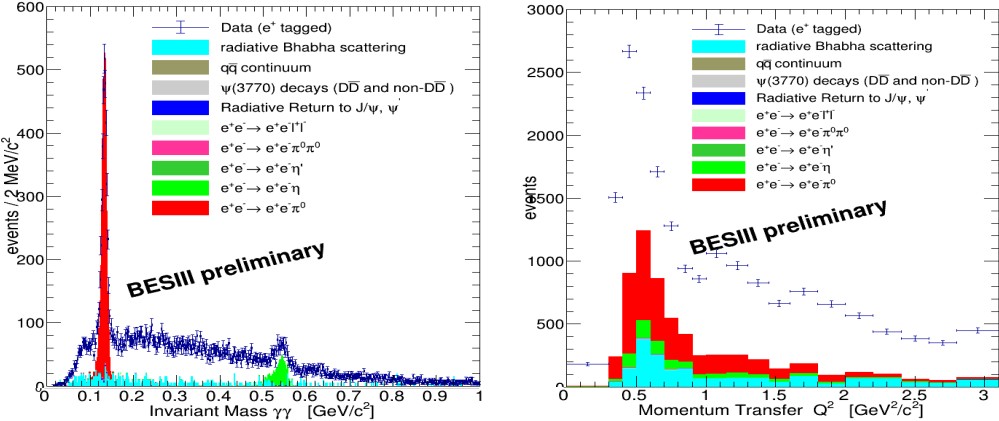

Figure 3: The $\gamma\gamma$ invariant mass distribution (left) and $Q^2$ distribution (right) from data and MC simulations. The dot with error bars are data, the red histogram is from signal MC simulation, other colored histograms are from background MC simulations.

calculated. The TFF as a function of $Q^2$ is extracted by dividing out the point like cross-section. The result is as shown in Fig. 4. The precision in $Q^2 < 1.5$ GeV$^2$ is unprecedented, in the $Q^2$ region above, the precision is compatible to the CLEO [22] result.

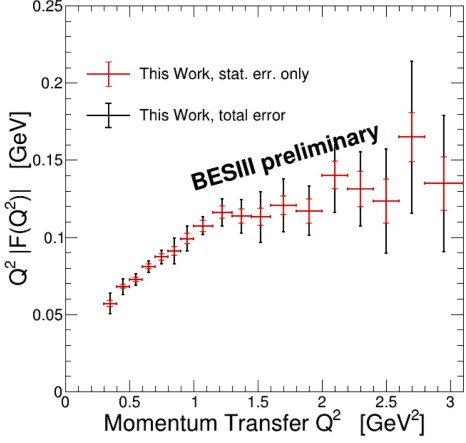

Figure 4: The preliminary result of the $\pi^0$ TFF from the BESIII experiment.

Comparing the TFF of $\pi^0$ measured from the BESIII experiment with the model calculations [26, 27] and the data-driven approaches [12, 28], the results are shown in Fig. 5. In the comparisons, the parameters from the model calculations or data-driven approaches are fixed according to the corresponding publications. A relative $\chi^2$, defined as $\sum_{i=1}^{\mathrm{nbin}=18} \frac{f_i^{\mathrm{exp.}} - f_i^{\mathrm{theo.}}}{\Delta f_i^{\mathrm{exp.}}}$, is used to obtain the goodness of the agreement. Here $f_i^{\mathrm{exp.}}$ is the TFF from the BESIII measurement, $f_i^{\mathrm{theo.}}$ is the value from the theoretical calculations, and $\Delta f_i^{\mathrm{exp.}}$ is the uncertainty of the TFF from the BESIII measurement. Among the comparisons to the model calculations, the 3−Octet model yields the smallest $\chi^2$, ($\chi^2 = 5.94$), 2−Octet model has the largest $\chi^2$ ($\chi^2 = 24.14$). The $\chi^2$ values for other models are around 9. Considering the uncertainty of the measurement, the descriptions from different models are compatible. The dispersively con-

structed TFF agrees with the measurement quite well within the uncertainties ($\chi^2 = 11.52$). However, the lower edge of the theoretical uncertainty band agrees with the measurement better. The description of the TFF using Padé approximant is model independent. It uses the TFF from previous measurements in both space-like and time-like region to determine the parameters. The comparison with the BESIII measurement shows very good agreement ($\chi^2 = 5.74$).

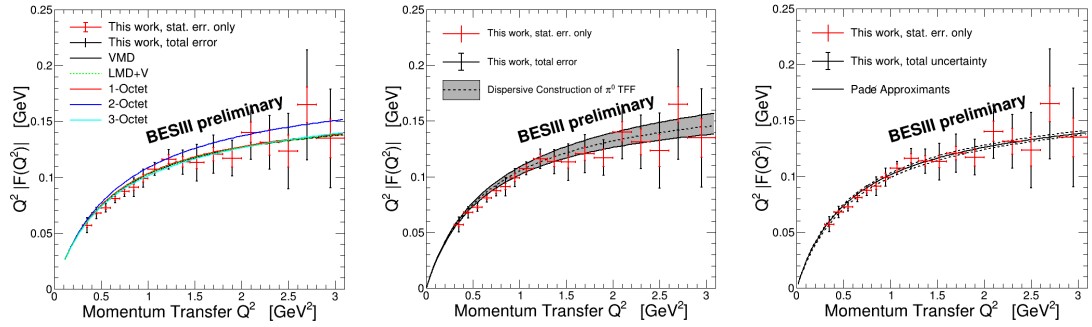

Figure 5: The comparison of the TFF of $\pi^0$ with model calculations (left), dispersive approach (middle), and Padé approximant (right). The dots with error bars are from the BESIII measurement, the curves with bands are from theoretical calculations.

With an analysis strategy similar to that used in the $\pi^0$ TFF measurement, the TFFs of $\eta$ and $\eta'$ in space-like regime are measured at BESIII experiment as well. The decay modes used are $\eta \rightarrow \pi^+\pi^-\pi^0$ and $\eta' \rightarrow \pi^+\pi^-\eta$, respectively. Both $\pi^0$ and $\eta$ are reconstructed by their decay into $\gamma\gamma$. The TFFs can be extracted in the region $0.3 \leq Q^2 \, [\text{GeV}^2] \leq 3.5$ with a precision comparable to the previous results from the CELLO [21] and CLEO [22] experiments but in a finer binning of $Q^2$. Adding more decay modes and including the data samples at CM energies above 4.0 GeV, the precision of these TFF measurements can be improved significantly.

## 4.2 Time-like transition form factor measurement of $\eta'$

Using $1.31 \times 10^9$ events taken at the $J/\psi$ peak, the TFF of $\eta'$ in time-like region has been measured at the BESIII experiment using Dalitz decay process $\eta' \rightarrow \gamma e^+ e^-$ [29]. It is the first measurement of the $\eta'$ Dalitz decay with an $e^+e^-$ pair in the final state. $864 \pm 36$ signal events has been found by fitting to the $\gamma e^+ e^-$ invariant mass distribution. The branching fraction $\mathcal{B}(\eta' \rightarrow \gamma e^+ e^-)$ has been determined to be $(4.69 \pm 0.20(\text{stat}) \pm 0.23(\text{sys})) \times 10^{-4}$. The transition form factor is extracted in eight $M_{e^+e^-}$ ($q$) bins from 0.1 GeV/$c^2$ to 0.8 GeV/$c^2$. The square of the TFF is fitted with a single pole parameterization:

$$|F(q^2)|^2 = \frac{\Lambda^2(\Lambda^2 + \gamma^2)}{(\Lambda^2 - q^2)^2 + \Lambda^2\gamma^2}, \tag{1}$$

where the parameters $\Lambda$ and $\gamma$ correspond to the mass and width of the Breit-Wigner shape for the effective contributing vector meson, and $q$ is the momentum transferred to the lepton pair. The fit result is shown in Fig. 6.

The $\Lambda$ and $\gamma$ values determined from the fit are $\Lambda_{\eta'} = (0.79 \pm 0.04 \pm 0.02)$ GeV, and $\gamma_{\eta'} = (0.13 \pm 0.06 \pm 0.03)$ GeV. The slope of the TFF corresponding to $(1.60 \pm 0.17 \pm 0.08)$ GeV$^{-2}$ and agrees within errors with the Vector Meson Dominance predictions and previous measurements.

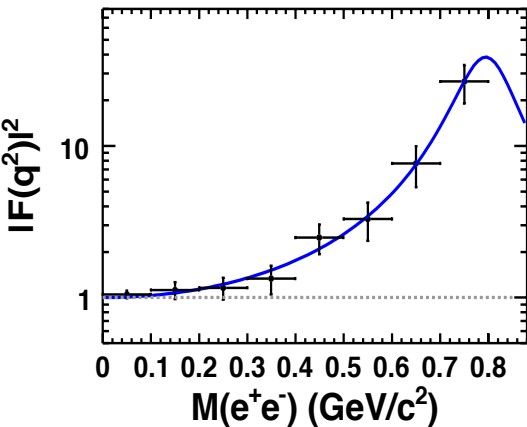

Figure 6: The TFF of $\eta'$ from the BESIII experiment using Dalitz decay process. The dot with error bars are the measurement and the blue curve is the fit result with the single pole approximation.

## 5  Measurement of $\gamma\gamma^* \to \pi^+\pi^-$

The contributions from meson loops, $\pi\pi$, $KK$, $\cdots$, are also important ones in the calculation of $a_\mu^{\mathrm{HLbL}}$. A dispersive analysis for these final states is needed due to the fact that the resonances in these final states have finite hadronic decay width, and there are non-resonant contributions. Dispersive approaches have been developed [10, 15, 16] recently, experimental measurements of $\gamma^{(*)}\gamma^{(*)} \to \pi\pi$ and $\gamma^{(*)}\gamma^{(*)} \to \pi\eta$ are important test for the validity of this approach.

The $\pi^+\pi^-$ final state was measured by the MarkII [30], CELLO [31] and Belle [32] experiments, but all in untag method. The cross section as a function of the invariant mass of $\pi^+\pi^-$ ($W$) from these measurements are shown in Fig. 7. The measurements from CELLO and Belle measurements start from $W > 0.8$ GeV/$c^2$. The only measurement at the $\pi^+\pi^-$ mass threshold region was done by the MarkII experiment with large uncertainties and a gap in the region between $0.4 - 0.7$ GeV/$c^2$.

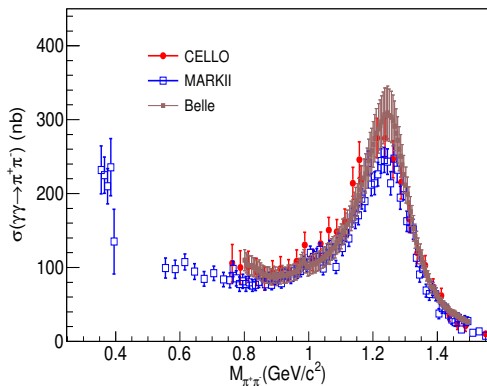

Figure 7: The cross section of $\gamma\gamma \to \pi^+\pi^-$ as a function of the invariant mass of $\pi^+\pi^-$ from the MarkII [30], CELLO [31], and Belle [32] experiments.

The study at the BESIII experiment is performed with single-tag method. The signal events are selected by requiring exact three charged tracks reconstructed in the detector. Two of

them are identified as pions, the remaining is taken as an electron or positron. The dominant background contributions come from $e^+e^- \to e^+e^-\mu^+\mu^-$ processes and $e^+e^- \to e^+e^-\pi^+\pi^-$ process (non two-photon process). The $e^+e^- \to e^+e^-\mu^+\mu^-$ background events is introduced because of $\pi$-$\mu$ misidentification. The cross-section is about 6 times larger than that of the signal process. This interaction is well-understood from the studies at the LEP. MC generators developed for the LEP energy scale [33, 34] have been validated in the BESIII energy region. Background contributions remaining after separating pions from muons with a multi-variable analysis are subtracted using MC simulations. Backgrounds with the same final states as the signal events are mainly from the radiative Bhabha scattering events, where the radiative photon couples to a vector meson, such as $\rho$ and $\omega$ in the case of $\pi^+\pi^-$ final state. These events peak in the $\pi^+\pi^-$ invariant mass spectrum and are subtracted by fitting to the $\pi^+\pi^-$ spectrum in bins of $Q^2$ and $\cos\theta^*$. Here $\cos\theta^*$ is the helicity angle of the $\pi$ in the CM frame of $\gamma\gamma$.

The remaining events are pure $\gamma\gamma^* \to \pi^+\pi^-$ events. From the $\pi^+\pi^-$ invariant mass spectrum, a clear $f_2(1270)$ signal is observed, as well as an accumulation of events in the $f_0(980)$ mass region. The clean signal sample allows a measurement of the differential cross-section in bins of $Q^2$, $W$, and $\cos\theta^*$. This is the first measurement of the two-photon $\pi^+\pi^-$ process with a single-tag method. The measurement can provide data points for $Q^2$ region from 0.1 GeV$^2$ to 4.0 GeV$^2$, $W$ from the $\pi^+\pi^-$ invariant mass threshold to 2.0 GeV/$c^2$, and a full $\cos\theta^*$ coverage $|\cos\theta^*| < 1.0$.

## 6 Conclusion

The experimental input for $a_\mu^{\text{HLbL}}$ calculation, including the TFF of the pseudoscalar mesons in both space-like region and time-like region, the helicity amplitude of the $\pi^+\pi^-$ final state have been studied at the BESIII experiment. These variables have been measured in the most relevant $Q^2$ region. The TFF of $\pi^0$ measured at BESIII is unprecedented in the $Q^2$ region from 0.3 GeV$^2$ to 1.5 GeV$^2$. The comparison between the experimental result and the theoretical calculations shows good agreement. The first single-tag $\gamma\gamma^* \to \pi^+\pi^-$ analysis can provide measurement in small $Q^2$ region, as well as in the low $\pi^+\pi^-$ invariant mass region down to the threshold with full coverage of $\cos\theta^*$. These measure are important inputs to the calculation of the HLbL contribution to $a_\mu$ using a dispersive approach.

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
