# Peer review of "Experimental input from $e^+e^-$ for $a_{\mu}$ light-by-light"

_SciPost Physics Proceedings, doi:SciPost Phys. Proc. 1, 054 (2019)_

## Round 1 · Referee Report · Gerco Onderwater · 2018-12-4

Report

This paper reports on important new measurements in their own right, and which may assist in reducing the uncertainty in the predicted value of the anomalous magnetic moment of the muon, specifically the hadronic light-by-light one. The theoretical motivation is sketched concisely, with a more elaborate description of the essence of the experimental procedure used to extract the various form factors.
Minor changes to the content, esp. grammar, would improve the already high quality of the paper.

Requested changes

L28: "with a statistical precision of 0.54 ppm" -> that is the total uncertainty
L34: "The QED contribution is the largest one, *AND* has been calculated up to 5-loop in perturbation theory with a precision of 0.0007 ppm" -> grammar
L35: "The weak contribution is small, *IT* has been calculated to 2-loop, with the measured Higgs mass taken into account [8], and its uncertainty is well under control." -> grammar; remove 'it'
L43: "... can be related to the hadronic cross-section ..." -> which hadronic cross section? There is no THE hadronic one.
L45: "*MEAN*while the situation for the HLbL part is different."
L72: "The BEPCII is a τ-charm factory THAT works with center-of-mass (CM) energy from 2.0
73 GeV to 4.6 GeV. "
L154-162: its not clear how to interpret the chi-squared, as the values are nowhere near 1. Some notion about the number of degrees of freedom is warranted, or a statement that only relative chi^2's are used.

  • validity: top
  • significance: top
  • originality: high
  • clarity: high
  • formatting: excellent
  • grammar: good

Author:  Yuping Guo  on 2018-12-05  [id 365]

(in reply to Report 1 by Gerco Onderwater on 2018-12-04)

Changes have been made according to the suggestions from referee

---

## Editorial Decision

published